# Vertical Farming Monitoring: How Does It Work and How Much Does It Cost?

**DOI:** 10.3390/s23073502

**Published:** 2023-03-27

**Authors:** Paula Morella, María Pilar Lambán, Jesús Royo, Juan Carlos Sánchez

**Affiliations:** 1TECNALIA, Member of BRTA (Basque Research Technology Alliance), 50018 Zaragoza, Spain; 2Department of Design and Manufacturing Engineering, University of Zaragoza, 50018 Zaragoza, Spain

**Keywords:** vertical farming, cost model, monitorization

## Abstract

Climate change, resource scarcity, and a growing world population are some of the problems facing traditional agriculture. For this reason, new cultivation systems are emerging, such as vertical farming. This is based on indoor cultivation, which is not affected by climatic conditions. However, vertical farming requires higher consumption of water and light, since in traditional agriculture those resources are free. Vertical cultivation requires the use of new technologies and sensors to reduce water and energy consumption and increase its efficiency. The sensorization of these systems makes it possible to monitor and evaluate their performance in real time. In addition, vertical farming faces economic uncertainty since its profitability has not been studied in depth. This article studies the most important variables when monitoring a vertical farming system and proposes the sensors to be used in the data acquisition system. In addition, this study presents a cost model for the installation of this type of system. This cost model is applied to a case study to evaluate the profitability of installing this type of infrastructure. The results obtained suggest that the investment made in VF installations could be profitable in a period of three to five years.

## 1. Introduction

The rapid increase in the world’s population poses a challenge for agricultural production. More and more people need to be supplied, with greater problems. These include resource scarcity, climate change, and a shrinking rural population. The agricultural sector is currently facing these challenges (growing population, resource scarcity, and climate change) with the help of precision agriculture and new technologies, such as Artificial Intelligence (AI), business intelligence, or Internet of Things (IoT).

The acquisition of data through these technologies and their subsequent analysis and evaluation makes it possible to increase production and optimize the resources consumed [1]. However, there is still a lack of adequate skills among farmers to apply precision agriculture [2]. In this sense, the acquisition of variables for production control could improve the understanding of precision agriculture, enhancing systems control (temperature, light, nutrition, etc.) through a friendly interface, as identified by Morella et al. (2022) in their literature review [3].

Regarding the acquisition of data, IoT (Internet of things) technology is considered as a key for optimizing production in agriculture [4]. IoT can be defined as a System of Systems (SoS) whose elements are independent, situated, and temporarily networked to synergically provide advanced cyber–physical functionalities [5]. As an example, Namani et al. (2020) introduced a smart drone for crop management. The smart drone used IoT and Cloud Computing technologies to develop a sustainable agriculture by controlling the cost, and monitoring performance and maintenance. In this work it is shown that real time acquisition of data by IoT and its processing and visualization through Cloud Computing help reduce costs, increase efficiency, and make smarter decisions [6].

Due to the possibilities offered by new technologies, new business models have emerged in the sector. Particularly, this paper is focused on Vertical Farming (VF). VF tries to overcome the horizontal agriculture to achieve more efficiency. VF was defined by Sharath Kumar et al. (2020) as “a multilayer indoor plant production system in which all growth factors, such as light, temperature, humidity, carbon dioxide concentration ([CO_2_]), water, and nutrients, are precisely controlled to produce high quantities of high-quality fresh produce year-round, completely independent of solar light and other outdoor conditions. This control can be fully automated by using sensors and imaging techniques in combination with crop simulation models and artificial intelligence” [7].

Currently, VF is expanding rapidly thanks to the increasement of consumer demand for sustainable, local, and fresh products, and the development of affordable lighting technologies [8]. Specific soil and climate conditions are not necessary for VF, so it enables the cultivation in disadvantaged climatic areas and reduces the soil pollution. In addition, the yield per area is higher than traditional agriculture. However, there are some disadvantages in comparison with traditional agriculture. Outside, in traditional agriculture, light, CO_2_ and water are free. Whereas in VF they must be assumed as cost [9]. Therefore, the use of new technologies in VF are crucial to minimize these costs and their environmental implication.

VF is introducing new technologies to face these challenges. For example, AI is being used for monitoring crop growth in VF increasing production, for example, color images are used to monitor crop growth [10]. In addition, AI can predict plant behavior [11]. Regarding the use of sensors and IoT technologies, they can be used for controlling and increasing the efficiency of water use, as can be seen in Ref. [12]. Moreover, Sivamani et al. (2013) proposed a combination of sensors and technologies for monitoring and automatically controlling VF [13]. However, as it is suggested in Saad et al. (2021), it is necessary for the combination of IoT, big data, and cloud computing technologies to develop a decision-making tool to improve, among others, productivity, quality, and costs [14]. Moreover, the financial uncertainty of VF is another challenge around this new agriculture [15]. There is not a specific cost model where all the costs, including sensors and IoT architecture, are considered.

Our research proposes a methodology to acquire the real-time data by sensors and IoT devices to show relevant information for VF farmers. Furthermore, we provide a detailed cost analysis of VF infrastructure and estimate the viability of VF implementation in a case study. Once the terminology has been conceptualized in this section, Section 2 proposes the methodology for acquiring data in real time and for estimating costs. After that, we present and discuss the results of the estimation. The paper ends with the conclusions and proposals for future studies.

## 2. Materials and Methods

Our methodology is divided into three steps. First, we analyzed what are the relevant variables for measuring VF production and what sensors can acquire those variables. After that, we describe the monitorization platform using IoT technology. Finally, the cost structure for VF infrastructure is presented.

### 2.1. Identifying Relevant Variables and Sensors for VF Monitoring

Analyzing previous studies about IoT implementation in VF [16,17], we have identified a common trend regarding used variables. Thus, the variables and Key Performance Indicators (KPIs) that can provide important information about the process are described below. Regarding carbon footprint calculation, a CO_2_ sensor and a network analyzer are required. Both variables, CO_2_ emissions and electricity consumption, will be used to calculate the carbon footprint.

Water footprint requires monitoring water consumption through a flowmeter. Humidity (both soil and ambient) inform us about the evaporation process, which must be included in water footprint calculation, and the water intake of the plants. When the humidity level is high, the stomata are open and absorb CO_2_, whereas if the humidity is low, the stomata tend to close. By measuring this variable, optimal water conditions for plant growth can be controlled. For example, an excess of humidity could lead to the presence of fungi in the plants, a fact that is very detrimental to the process, so it should be kept within the appropriate limits for crop growth [18]. Another necessary variable for measuring water footprint is ambient temperature. Each type of crop requires different temperature conditions for the different phases of its growth. Excessive temperature can damage the leaves of the crop and, therefore, the development of the plant. Therefore, a humidity and temperature sensor is required for measuring both variables in VF installations. In addition, we need a soil humidity sensor for specifically measuring the soil humidity.

Furthermore, there are three eco-physiological properties that are measured:Crop growth: measured by the CO_2_ and O_2_ levels. The CO_2_ level is essential for plant photosynthesis to occur and for the correct development of the crop [19]. The O_2_ level is very relevant during the growth process of crops, because, if it is not in the right range, the plants receive neither enough oxygen nor water and therefore, they can no longer absorb nutrients properly, ending in crop death due to the lack of nutrition [19];Evapotranspiration: crop water requirements decrease and therefore, water management is more efficient since water is only supplied when the plant requires it [18]. Evapotranspiration is measured by air flow and vapor pressure. Air flow can affect crop development (it can damage leaves, break branches, soil erosion, etc.) and hinder the application of agrochemicals. Wind has two important variables: direction and speed, and it can be dry, cold, wet, etc. [20]. In addition, it is directly related to the pressure and, therefore, to the admission of CO_2_ by the leaves [18]. Regarding vapor pressure, air vapor pressure must be lower than vapor pressure inside the stomatal cavity of the plant to drive transpiration. This fact is crucial for photosynthesis to develop properly; therefore, this variable must be within the appropriate range to ensure good CO_2_ uptake. A wind speed sensor and a steam pressure sensor are necessary;Light characteristics: monitoring the light characteristics is crucial for the process because the use of light-emitting diodes (LEDs) as light sources can initiate and maintain photosynthesis reactions—just as sunlight does [21]. The measured light variables to improve crop growth are optical wavelength, illuminance (lux), and irradiance intervals. Illuminance is the amount of luminous flux that a surface receives per unit area, assumed to be uniformly illuminated. Regarding wavelength range, due to the eco-physiological aspects of crops, light in the blue range is usually needed in the initial phase of the growth process, while red light is required in the final phase [22]. In Figure 1 one can see in more detail the effect of wavelength on the growth process of the plant. This radiation is expressed as synthetic photon flux density, i.e., the total number of photons between 400 and 700 nm per surface area and time (μmol/m^2^/s). Finally, the intensity of photosynthetically active radiation (PAR), that is, the fraction of light in the radiation, ranges from 400 to 700 nm, which drives plant photosynthesis. These characteristics are measured by a silicon pyranometer smart sensor.

Chlorophyll content: chlorophyll is a green pigment found in plant leaves. It is a photoreceptor, i.e., it captures light and uses it to perform photosynthesis and create both sugars and nutrients that feed the plant. Therefore, chlorophyll content is directly related to nutrients (nitrogen) and therefore it is important to know its content for proper crop growth [21]. So, a chlorophyll content sensor is crucial to know if the crops are growing well.

All the KPIs and variables are summarized in Table 1.

### 2.2. IoT and Data Acquisition Arquitecture

It is necessary to develop a data acquisition system based on IoT because the key to VF is monitoring and controlling the cultivation from the outside. This system can be used in agricultural environments to collect data on the variables presented in Section 2.1.

This system will be distributed, wireless, and based on measurement nodes. In addition, it will be able to connect remotely, in case there is a network connection, and to operate with a local server.

The data obtained from the sensors presented in the previous section are captured through an IoT device. This device is a microcontroller with WIFI capability and is prepared for hostile environments. Additionally, it is reprogrammable according to the requirements of the process and has the possibility of integrating long distance communications through specific modules. Sensors acquire the data in a period time of 15 min and the IoT device sends the information to the cloud in a period time of 30 min. Therefore, the visualized information is refreshed every 30 min.

As a result, the IoT architecture in VF is presented as follow (see Figure 2).

### 2.3. Cost Structure

Regarding the costs associated with the VF infrastructure, no approach has been detected in the literature review that shows how to calculate the cost of production in this new VF system.

After reviewing and investigating these new VF systems, studying the operations, needs, and requirements of the production process, this work proposes a model for calculating the cost of implementing this type of farming (see Figure 3).

As shown in the diagram above, the total cost of the infrastructure (C_VF_) is calculated by adding the following terms (Equation (1)):C_VF_ = C_raw material_ + C_infrastructure_(1)

The calculation of each block is detailed below.

Cost of raw materials (C_raw material_) refers to the acquisition cost of those elements from which the consumer products are obtained; in this case they will be (Equation (2)):C_raw material_ = C_crop_ + C_pollinators_(2)

The cost of the infrastructure (C_infrastructure_) includes all those facilities, systems, and technologies that are used and are necessary in the VF cultivation process.

The cost of all the infrastructure involved in the process can be defined by the following expression (Equation (3)):C_infrastructure_ = C_installations_ + C_sist&tech_ + C_renewable energy systems_(3)

The cost of installations includes all those physical facilities that are necessary for production (water, artificial lighting, etc.) and the furniture necessary to house the crops. Thus, the cost of the facilities will be (Equation (4)):C_installations_ = C_container_ + C_shelves_ + C_floor_ + C_systems water & lighting_(4)

The cost of systems and technologies (C_sist&tech_) includes those technologies necessary for the measurement processes of the variables, their control, handling, and transport of the crops within the production process, etc.

Therefore, the expression for the calculation of this block is (see Equation (5)):C_sist&tech_ = C_sensors_ + C_software_ + C_handling/support_(5)

The cost of the sensors will consider the cost associated with their installation and the unit cost per number of sensors, as shown in the following Equation (6):(6)$$C_{\mathrm{sensors}}=\sum_{0}^{n}{\left(C_{\mathrm{installation}}{+C}_{\mathrm{software}}\right)}_{\mathrm{sensor}\;i}$$

As for the cost of the software (Equation (7)) to carry out all these control tasks during the production process, it must include the acquisition of the program, the cost of the installation contract, the cost of possible employee training for its proper use, the cost of possible improvements that could be introduced in the software, etc.
C_software_ = C_program_ + C_installation_ + C_formation_ + C_improvements_(7)

Finally, the cost of the handling and support block, which includes the robots and AGVs inside the container, is calculated as a material resource belonging to the company, as in the case of the container.

For its calculation, the acquisition cost, the corresponding amortization cost for the use of the resource and the possible installments of financing payments made for the payment of the resource at the time of acquisition must be considered.

Regarding the cost of renewable energy systems, this cost must include everything related to the installation (modules, supports, storage, etc.) and amortization. Thus, the expression for the calculation of the renewable energy system is (see Equation 8):C_renewable energy systems_ = C_installation_ + C_amortization_ + C_maintenance_(8)

To sum up, this cost model provides farmers an accurate estimation of the infrastructure costs, including the implementation of new technologies. Thus, they can evaluate the investment viability.

## 3. Results

This section presents the results of applying the VF system in a particular case study. First, we present and characterize the case study. Then, we calculate the cost model. Finally, we discuss the results obtained using investment viability indicators.

### 3.1. Case Study Definition

The proposed case study aims to grow cherries in a VF system, so that cherries can be obtained outside of their usual growing date. The VF system is located in Zaragoza, Spain.

To do so, it employs a 40-ft container (67.7 m^3^ volume) to house the VF system. Inside the container there are fixed racks (model RADIX from SANANBIO [23]) for the placement of the trees. Considering the usable area of the container and the dimensions of the racks, it is possible to install 16 racks distributed in 2 rows of 8. As for the floor, it is assumed that the container is installed on land already purchased by the company.

We will discuss some important features regarding the water and lighting systems necessary for irrigation and crop growth. The design proposes the installation of 17 valves (HUNTER PGV 24 V), installing a general valve for opening and closing the irrigation, and one on each rack in case it is necessary to intervene in any of them due to maintenance problems. In addition, drippers will be installed to supply water to each rack. As for the cost associated with the lighting fixtures needed to grow the crops, a total of 48 fixtures are needed (one LED strip for each tray).

Regarding IoT architecture, below is a table (Table 2) showing the number of sensors of each type to be installed. It should be noted that the measurement of soil moisture, illumination, and chlorophyll content will be carried out only in one tray of each shelf at different heights to observe if there are any possible variations in the trays.

To conclude with the systems and technologies block, it is proposed that the handling and support systems will be robots and AGVs inside the container.

Regarding the installation of renewable energy systems, it is necessary to know the energy consumption of the container, the available surface to install the panels, and the orientation and inclination.

In terms of energy consumption, two situations will be differentiated: winter and summer. It is important to note that the average consumption of a 40-ft container is 7.8 kWh working in steady state, i.e., when the container reaches the desired temperature, it stops automatically. Thus, its consumption will depend largely on the ambient temperature outside, so some factors will be included to calculate the consumption in the different situations [34]. Adding to these calculations, the consumption of LED luminaires is 3749.76 kWh in summer and 3548.16 kWh in winter, and the total consumption is 28,273.68 kWh. Therefore, based on this consumption and the peak power of the panels, the number of modules required for the energy supply is calculated.

The panel model chosen is the KuMax CS3U-360P from Canadian Solar. Thus, using the RetScreen program for sizing photovoltaic installations, it is concluded that with 62 panels of 360 Wp with an inclination of 35° and facing south, it is sufficient to cover the consumption of the study container. The annual electricity generation of the installation will be almost 29 MWh per year, i.e., higher than the consumption, and it would occupy about 100 m^2^ of land.

### 3.2. Cost Model Calculation

The table below (Table 3) shows the cost of each part of the VF system that has been applied into the equations defined in Section 2.

As can be seen in Table 3, the total cost of implementing a VF infrastructure for growing cherries is estimated to be almost EUR 140,000. The viability of this inversion is evaluated in the following section.

## 4. Discussion

Crops of stone fruit in VF are not a widespread production process. For this reason, the price at which the fruit, in this case cherries, could be sold out of season is unknown, although it is estimated that it could be double its usual price.

As the real profit that can be obtained from this process is still unknown, ROI (Return of Investment) can be not calculated. Therefore, to analyze the results of our case study, we have used PAYBACK, reformulating the equation (see Equation (9)). Thus, we can analyze the income that a company would require to recover the investment between 1 and 10 years (see Figure 4).
Income = Investment/number of months(9)

As can be seen in Figure 4, the necessary income decreases rapidly from year 1 to year 2 and it can be considered stable from year 5. Since a new infrastructure is a large investment, it can be considered as valid to recover the investment between three and five years. This seems possible with the estimated income.

## 5. Conclusions

This study proposes a set of sensors and an IoT architecture to collect variables in real time. In addition, a cost model of the infrastructure required to make a vertical farming implementing new technologies is presented.

This is intended to provide an answer to the problems that farmers may have when adapting to new technologies. A data acquisition architecture is offered, which allows them to collect information in a simple way and in real time on the state of the crop. In addition, the cost model and the case study presented provide an estimate of the cost of installing a new VF infrastructure with new technologies implemented in it. This case study shows that the payback of the investment in these systems can be between three and five years. In this way, it is intended to eliminate the barriers and prejudices that may exist towards this new type of crop.

As future research, actions are proposed to help the farmer improve the vertical cultivation process and to eliminate the barriers to the implementation of these systems; for example, by the development of more complex indicators based on the variables presented in this article. Regarding costs, the aim is to continue developing the cost model until the cost of growing stone fruit in FV can be calculated. Finally, we will identify and propose actions to improve the sustainability of this process, such as the development of renewable energy systems and the reuse and control of water consumption, among others.

## Figures and Tables

**Figure 1 sensors-23-03502-f001:**
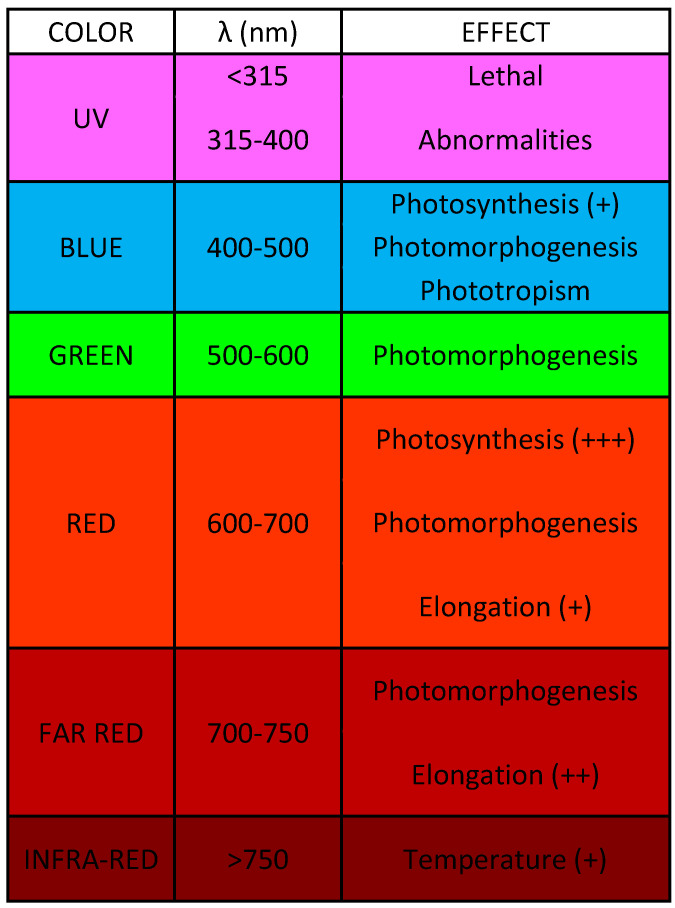
Effect of wavelength on the growth process of the plants. The colors of the figure represents the wavelength colors. Furthermore +, ++ and +++ means positive effect, very positive effect and excellent effect, respectively.

**Figure 2 sensors-23-03502-f002:**
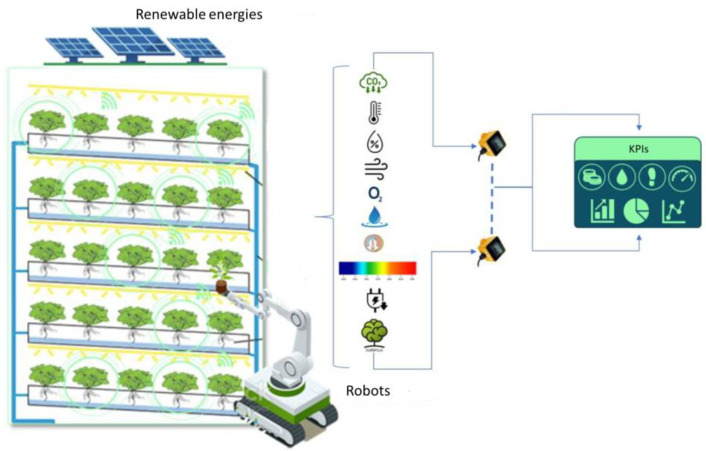
The IoT architecture.

**Figure 3 sensors-23-03502-f003:**
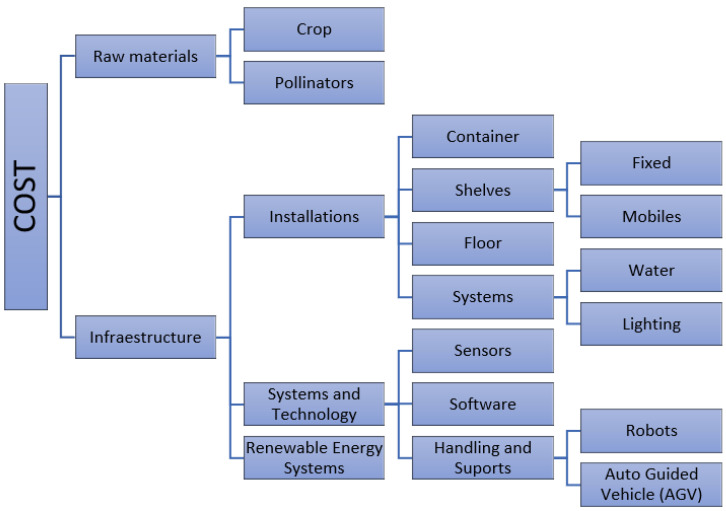
The cost structure.

**Figure 4 sensors-23-03502-f004:**
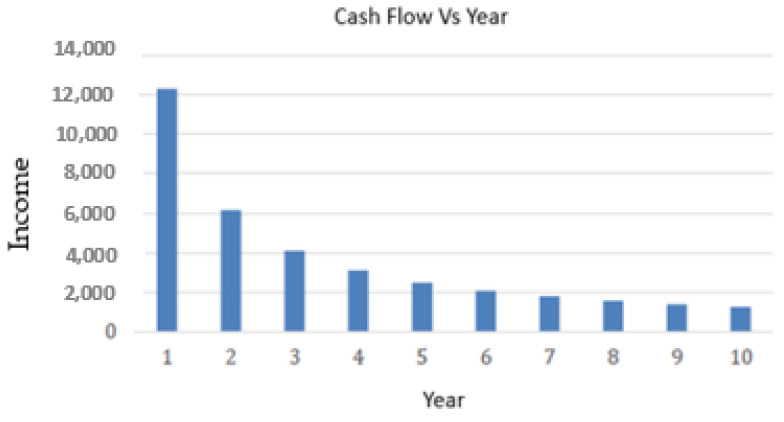
Income vs. year.

**Table 1 sensors-23-03502-t001:** Relation between the variables and indicators.

Key Performance Indicator		Variable
Water footprint		Humidity
		Temperature
		Water consumption
Carbon footprint		CO_2_
		Energy consumption
Eco-physiological properties	Crop growth	O_2_
		CO_2_
	Chlorophyll content	Chlorophyll
	Light characteristics	Optical wavelength
		Illuminance
		Irradiance
		PAR
	Evapotranspiration	Vapor pressure
		Wind speed and direction

**Table 2 sensors-23-03502-t002:** Cost of the sensors. €—EUR.

Sensor	Nº of Sensors	Unit Cost	Total
IoT device [24]	1	61.17 €	61.17 €
Temperature and humidity sensor [25]	2	31.96 €	63.92 €
Soil humidity sensor [26]	16	8.00 €	128.00 €
Wind speed sensor	2	50.00 €	100.00 €
CO_2_ sensor [27]	2	434.95 €	869.90 €
O_2_ sensor [28]	2	356.95 €	713.90 €
Flowmeter [29]	1	155.00 €	155.00 €
PASCO Wireless Light [30]	16	421.45 €	6743.20 €
Steam pressure sensor [31]	2	430.76 €	861.52 €
Chlorophyll content sensor [32]	16	2257.86 €	36,125.76 €
Network analyzer [33]	1	281.94 €	281.94 €
Network and connection system	1	1960.00 €	1960.00 €
Total C_acquisition sensors_	48,064.31 €		

**Table 3 sensors-23-03502-t003:** Values applied in the cost model structure. €—EUR.

Cost Model Structure					
C_installations_	C_container_	Acquisition [35]		8000 €	8270.00 €
		Installation	Time	3 h	
			Labor cost (four operators)	10 €/h	
			Crane rent	150 €	
	C_shelves_	Acquisition + installation (16 shelves) [36]		218 €	3488.00 €
	C_floor_	0 €			0 €
	C_systemswater_	Installation	Time	32 h	1294.00 €
			Labor cost (four operators)	10 €/h	
		Conducts [37]	15 €		15 €
		Valves [38]	19 €	17 valves	323.00 €
		Drippers [39]	0.39 €	16 drippers	6.24 €
	C_systemslighting_	Installation	Time	80 h	4044.00 €
			Labor cost (five operators)	10 €/h	
			Materials		27.00 €
		Conducts [37]			16.00 €
		Lighting	124.90 €	48 lighting fixtures	5995.00 €
C_sist&tech_	C_Sensors_	Installation	Time	32 h	1294.00 €
			Labor cost (four operators)	10 €/h	
		Acquisition	See Table 2		48,064.31 €
	C_Software_	32,000.00 €			
	C_Handling and supports_ [40,41]			12,400.00 €	
C_renewable energy systems_ [42]				21,215.48	
C_total_		138,452.00 €			

## Data Availability

Not applicable.

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
