# Peer review of "Vertical Farming Monitoring: How Does It Work and How Much Does It Cost?"

_sensors, 2023, doi:10.3390/s23073502_

Round 1

Reviewer 1 Report

Hello,

page 9 (289) figure 3 means figure 4, in the following text (290) it is figure 2?

The equation 9 is not usable for cash flow. What is the income of the VF?

The income ist not calculated. The price for cherry is unknown?

What's abaout the yield? Is the system  designed for cherry trees (figure 2)? The figure 4 cash flow is just hypothesis.

ROI can be not calculated (283). 

292 ....it can be considered as valid to recover the investment between 3 - 5 years??? 

The results are not clear.

Author Response

Please, find attached our answer.

Reviewer 2 Report

Improving yield quality and stability is an ongoing challenge for agriculture and breeding programs. Climate and soil conditions are variables that severely hamper yield prediction. The authors note this problem and undertake to explore an alternative type of conventional or organic farming, vertical farming. This is an innovative approach. In addition to the topic of farming without controlling climatic conditions, the paper addresses the profitability of implementing vertical farming.  The conclusion in the work is the profitability of this agriculture after 3-5 years despite the consumption of light and water. 

I rate the work as very innovative and necessary for the development of the discipline of agriculture. 

The introduction is extensive confirmed by well-chosen literature. The material and methods are consistent. Very well presented graphs. Very clear tables. In the cost calculations, the authors explain in simple terms the meaning of each algorithm. Results are confirmation of the research hypothesis. The discussion is very pertinent. In the conclusion, the authors themselves state that this is an initial work, which I hope will give rise to new interesting results, which will consequently affect the development of agriculture. 

Author Response

Please, find attached our answer.

Reviewer 3 Report

Summary

The authors present an analysis on the sensors needed for vertical farming, an IoT system that can be used to acquire the data and a cost calculation for a case study implementation.

Major comments

I find the article very interesting, but further work is needed to produce a publication quality manuscript. The methods section lacks clarity and information. The results section misses on important parts of the study. A description of the IoT system is missing, and a reason/comparison of the IoT system versus other options is missing.

The writing of the manuscript is unprecise in many instances. There are 1, 2, and 3 sentence’s paragraphs that can be combined into a single coherent paragraph as they discuss the same idea. There is an excessive use of unnecessary connectors. Please carefully revise the writing of the entire manuscript to improve its readability and quality using the specific comments to identify issues and fix them thoroughly.

In the methods section the information is scattered. It is mentioned that a variable is “very important”, then another is “crucial”, the next one is “key”, other is “very relevant” for the footprint calculation. I think starting the section explaining how the footprint is calculated would tell the reader what are the variables needed. It is hard to gauge what is intended by “very relevant”, either a variable is needed to calculate the footprint, or it isn’t. Are there variables that have a higher importance? If this is the case, please explain why and how in precise terms.

The information that is presented in section 2.1 can be organized in a table, or a diagram rather than being dispersed all over the section. What are the variables that need to me measured? This can be a table. How are they relevant to the process? This can be explained in text supported by a diagram that shows the entire process.

Section 2.2 needs to be expanded to include actual information about the system. What is the point of this section? “Data is transmitted over Wi-Fi to an IoT device with a microcontroller” summarizes all the information presented in section 2.2. However, I think there is more information to be added, how do sensors transmit data (temporal resolution, frequency, format)? What is the microcontroller doing? There are more important details that can be added to this section to make it relevant. Further work is needed here.  

Section 2.3 needs to be reorganized to tell a coherent story. I don’t understand what is being discussed here. Are all these equations needed? Can the cost information be summarized somehow with a diagram supporting Figure 3? It is hard to follow this section as information is unnecessarily scattered. If mentioning that the cost of sensors includes software and support, then why is an equation needed to sum these terms? Simply mentioning the costs associated with each element will suffice.

Results section. The methods section is divided in 3 subsections, sensors, IoT, and cost. The results jump directly to cost. I understand you did not implement an actual case study, is this correct? Since costs vary by city, country, time of year… a discussion about that were the assumptions to arrive at each cost is guaranteed. What about a description of the IoT system? What about a comparison, in terms of functioning, of a non-VF system to produce similar yield?

I understand you do not know the price at which the fruit can be sold. However, you do know the price at which the fruit is sold, can this be used to provide a simple feasibility analysis?

Are table 3 and Figure 3 presenting the exact same information? Then why are both included? What are the implications of this result? Please review your manuscript carefully before resubmitting it. 

Minor comments

L10. I suggest “Vertical farming requires…”

L11. What factors are you referring to?

L12. I suggest removing “in this sense” and defining what wastes are you referring to.

L25. I suggest removing “, but”.

L26-28. What reason and what challenges are you referring to? I suggest removing “For this reason”.

L29-31. What technologies are you referring to? What data? What resources? I suggest removing “Thus”.

I will not make any additional specific comment regarding connectors. Please revise the rest of the manuscript and correct the excessive use of unnecessary connectors (e.g., moreover, however, in this sense, all things considered). The excessive use of connectors makes the article difficult to read, and in most cases the connectors are not conveying any additional or useful information. I found several instance where the same connector is used twice in the same sentence. Please revise thoroughly.

L32-34. Can you explain how the acquisition of variables can help understand precision agriculture and assist in overcoming the lack of farmer’s skills?

L39-43. Can you mention specifically what was done with the drone? I understand a drone was used and this helped, but how?

L44-46. I don’t understand the main idea of this paragraph. Should this text be a paragraph? Can you please expand or remove it, if not needed?

L55. Is a comma needed before “and the development…”?

L56-57. Please revise this sentence.

L63-65. Can you please explain how is AI being used to monitor crop growth? When mentioning “Also, it can predict” what is it referring to, and how is it doing it? Is “agrilcultures” a typo?

L65-66. This paragraph starts with “VF agrilcultures are introducing new technologies”, and this sentence is mentioning the same thing again. I suggest rephrasing or removing.

L67-69.  Please check MDPI format for in-text citations. “As it is suggested in [13]” probably needs to include the names of the authors. Check in the rest of the manuscript and format accordingly.

L71. “among others” is incomplete.

L72-74, Should this text be a paragraph?

L75. What is meant by “all things”?

L75-82. This could be a single paragraph.

L84. This should say “what are the relevant variables”.

L90. You identified a common trend regarding variables. This does not require a citation unless you mention the variables, otherwise is just saying you did something and citing two references for it. I suggest mentioning the variables here.

L104-106. Is this a paragraph? Also, a temperature and humidity sensor is needed or temperature and humidity sensors are needed.

L107. Please combine with other text to form paragraphs conveying ideas. A sentence saying a sensor is needed does not need to be a paragraph. Please revise the rest of the manuscript as there are many very short paragraphs that can be combined into a single paragraph (they convey the same idea). Also, it should say that both sensors are necessary instead of both sensors is necessary, please fix and revise also in the rest of the manuscript.

L112. What is meant by “all this”?

L 140. What is the source of this figure?  Is the figure relevant for a particular reason? When describing the other variables selected, no additional information about how they impact the growth process, why is this table here?

150. “Airflow is important to measure this variable”, please fix.

L160. Why is necessary to develop a data acquisition system based on IoT? Is this the only option?

AgAfter finishing section 2.1 I don’t know how many sensors are needed and how many variables are being monitored. A table would make this section much easier to follow.

ain, it is hard to follow the text with these very short paragraphs that don’t convey an idea, and are followed by more short paragraphs on the same topic.

Figure 3. What is AGV? What type of robots are you quoting here? Most of the terms presented in the cost structure have not been introduced and are not explained anywhere in the text. Please take the reader from point A: a description of the system, the elements involved and their function and then discuss the cost associated with each element.

L223. Is that a paragraph?

L229. “Talking about water and light” Please be precise and formal.

I have not reviewed the results, discussion, and conclusions with detail as there are too many issues in the methods and content. 

Please revise.

Author Response

Please, find attached our answer.

Round 2

Reviewer 1 Report

I have no more suggestions the paper is ok for publication.